# Combined NMR Spectroscopy and Quantum-Chemical Calculations in Fluorescent 1,2,3-Triazole-4-carboxylic Acids Fine Structures Analysis

**DOI:** 10.3390/ijms24108947

**Published:** 2023-05-18

**Authors:** Nikita E. Safronov, Irena P. Kostova, Mauricio Alcolea Palafox, Nataliya P. Belskaya

**Affiliations:** 1Department of Technology for Organic Synthesis, Ural Federal University, 19 Mira Str., Yekaterinburg 620002, Russia; safronov2401@yandex.ru (N.E.S.); n.p.belskaya@urfu.ru (N.P.B.); 2Department of Chemistry, Faculty of Pharmacy, Medical University—Sofia, 2 Dunav Str., 1000 Sofia, Bulgaria; 3Departamento de Química Física, Facultad de Ciencias Químicas, Universidad Complutense, 28040 Madrid, Spain; alcolea@ucm.es

**Keywords:** 1,2,3-triazoles, carboxylic acid, fluorescence, TDDFT, dimer, p*K*_a_

## Abstract

The peculiarities of the optical properties of 2-aryl-1,2,3-triazole acids and their sodium salts were investigated in different solvents (1,4-dioxane, dimethyl sulfoxide DMSO, methanol MeOH) and in mixtures with water. The results were discussed in terms of the molecular structure formed by inter- and intramolecular noncovalent interactions (NCIs) and their ability to ionize in anions. Theoretical calculations using the Time-Dependent Density Functional Theory (TDDFT) were carried out in different solvents to support the results. In polar and nonpolar solvents (DMSO, 1,4-dioxane), fluorescence was provided by strong neutral associates. Protic MeOH can weaken the acid molecules’ association, forming other fluorescent species. The fluorescent species in water exhibited similar optical characteristics to those of triazole salts; therefore, their anionic character can be assumed. Experimental ^1^H and ^13^C-NMR spectra were compared to their corresponding calculated spectra using the Gauge-Independent Atomic Orbital (GIAO) method and several relationships were established. All these findings showed that the obtained photophysical properties of the 2-aryl-1,2,3-triazole acids noticeably depend on the environment and, therefore, are good candidates as sensors for the identification of analytes with labile protons.

## 1. Introduction

The 1,2,3-triazole cycle is an important fragment in the molecules of various drugs [1,2,3], biologically active compounds [4,5,6,7,8,9,10,11], and polymers [12]. Its good accessibility, ease of modification, and specific structural properties, such as a high-dipole moment, molecular rigidity, and the ability to bind by weak intermolecular interactions (hydrogen bonds, dipole–dipole, and π-stacking) with a molecule target, as well as its promising photochemical properties, are attractive properties of triazoles. This has made them a focus of attention in multiple areas of science: chemistry, biology, medicine, physics, and material chemistry [2,13,14,15,16,17,18,19]. However, the functional groups and substituents surrounding the heterocyclic core are also very important, since they are effective tools for the design of new fluorophores and biologically active compounds [1,2,20,21,22,23]. Therefore, the issues in designing new 1,2,3-triazoles in accordance with the requirements of a specific direction of their use are acute and relevant at present.

Traditional approaches to the design of new fluorophores include many factors that have long been studied and are well-known. Among them, the most significant and widely used are the change in the nature of substituents and their localization with the formation of the A-π-D system, which is more sensitive to the microenvironment and analytes [24]. These factors are undoubtedly important and provide answers to many essential questions concerning the manifestation of photophysical effects and some other effects. However, in recent years, it has been found that, in real systems (solutions or solids), fluorescence is often provided by a more complex mixture of particles than individual molecules, and by a balance between fluorescent and nonfluorescent particles [25]. When fluorophores are aggregated, they can exhibit the same properties as their individual molecules. However, in most cases, they exhibit a partial or complete quenching of emissions, which is known as the aggregation-caused quenching (ACQ) effect, or initiate/increase their emission, which is termed aggregation-induced emission (AIE) or aggregation-induced emission enhancement (AIEE) [25]. Thus, heterocycles and the surrounding substituents are very important from the perspective of the emergence of new prospects for the design of fluorophores as tools for tuning intra- and intermolecular weak interactions. The nature and strength of these prospects determine the properties and stability of the supramolecular structures [22,23]. Thus, the precise knowledge of non-covalent interactions (NCI) is a crucial factor in the understanding of bio- and photoactivity, as well as a powerful tool for tuning these properties in accordance with possible applications [25,26,27,28,29].

Previously, we developed a simple, versatile, and effective one-pot approach for the synthesis of high-functionalized 2-aryl-1,2,3-triazoles that could be used to obtain several series of these compounds with different combinations of substituents and functional groups, and to study their photophysical properties [5,12,16,17,18]. Spectral investigations indicated high blue fluorescence with quantum yields (QYs) ranging from 10 to 99%, and lifetime variations (from 0.691 to 6.884 ns), as well as the unique ability of the substituents to control the photophysical characteristics and their sensitivity to the microenvironment and external factors. 2-Aryl-1,2,3-triazoles, depending on their structure, demonstrated a significant AIEE effect and, at the same time, a sharp decrease in the quantum yield in binary mixtures with insignificant changes in their structure [17,18]. An outstanding ability to selectively detect Hg^2+^ ions was revealed in some 1,2,3-triazoles with carboxamide groups [18]. The obtained results showed an increased sensitivity to both structural changes and the microenvironment, achieved by the drastic changes in QYs.

In the triazole compounds, the organic acids that possess electron-donating and electron-accepting centers in the molecule are a subject of special attention [9,11,17]. In a previous work [17], we showed the promising photophysical properties of a large set of newly synthesized 2-aryltriazolecarboxylic acids containing amino-groups. They demonstrated different behaviors, with a varying pH and drastic differences in p*K*_a_. It is known that organic acids can provide a stable associate (dimer, catemer, and bridged synthons, etc.) in solution using different non-covalent interactions [30,31,32,33]. Since the presence of a carboxyl group causes molecules to form various aggregates, in a continuation of our work on the design of a new effective fluorophore based on a 1,2,3-triazole core, we should evaluate the extent of their ability to form NCIs [34,35,36,37,38].

The aim of our current study was to invesitage the NCIs of 1,2,3-triazole acids (Figure 1) in various solvents and evaluate the influence of their molecular architecture on optical properties and biological potential. To better understand the nature of these NCIs and evaluate their influence on their properties, an extensive investigation was carried out for two representative examples of acids **1a**,**b**, and their sodium salts **2a**,**b** (as an ionized form of the acids).

The NCI strengths found between molecules’ active centers are highly dependent on the solvent that was used and the mechanism of solute–solvent interactions [39,40]. Therefore, to perform spectral studies and theoretic investigations, the solvents showing crucial differences in their mechanism of the interaction with solute were chosen. The formation of molecular aggregates in solution is a specific and complex phenomenon and requires a careful consideration of the equilibrium between the dissolution ability and the solvent’s ability to interact with the organic molecules, their excimers/supramolecular aggregates. For the investigation, experimental (UV- and fluorescence spectra, ^1^H, ^13^C NMR spectra) and theoretic (TD-DFT quantum chemical calculation) approaches were used.

## 2. Results and Discussion

### 2.1. Synthesis

2-Aryltriazolic acids (ATAs) **1a**,**b** were obtained by hydrolysis of the 5-amino-2-aryl-1,2,3-triazol-4-carbonitriles **3a**,**b**, as described in our previous paper [17]. The acids were then transformed to the sodium salts of 2-aryl-1,2,3-triazoles **2a**,**b** by treatment with NaOH (Figure 2).

The structure and purity of the ATAs **1a**,**b** and their salts **2a**,**b** were confirmed by complex spectral data and elemental analyses (Appendix A).

To characterize the acidic properties of compounds **1a** and **1b**, we determined the logarithm of the acid dissociation constant (p*K*_a_) [41,42,43,44,45,46,47]. The calculation was based on the ratio of the solution concentration and pH [46] and is presented in detail in the Appendix A. Experimental p*K*_a_ values for acids **1a** and **1b** in the mixture of H_2_O-DMSO (*v*/*v* = 9/1) were found to be 8.5 ± 0.1 and 7.3 ± 0.2, respectively (Appendix A). Thus, 2-aryl-1,2,3-triazole acids **1a** and **1b** can be classified as weak acids, and weaker than benzoic acid (p*K*_a_ = 4.2). At pH values of ≤ p*K*_a_ − 2, the substance is said to be fully protonated, and at pH ≥ p*K*_a_ + 2, it is considered to be fully dissociated (deprotonated). Thus, we can assume that these acids exist as a mixture of protonated/deprotonated forms at pH = 7, where **1b** is more ionized.

### 2.2. Photophysical Properties of the ATAs and Sodium Salts

#### 2.2.1. Experimental and Calculated Characteristics in Three Different Solvents

The absorption and fluorescence spectra of ATAs **1a**,**b** (Figure 1) are recorded in polar (DMSO), nonpolar (1,4-dioxane), and polar protic (methanol) solvents. The optical properties of salts **2a**,**b** were measured in DMSO and water, in accordance with their solubility (Table 1 and Figure 2). The absorption maxima in DMSO, 1,4-dioxane, and MeOH were in the 325–342 nm range. Acid **1a** showed a similar absorption maxima in DMSO and 1,4-dioxane, Δλ = 6.0 nm (Δν = 529 cm^−1^), whereas the **1b** absorption maxima were the same in these solvents. However, the optical characteristics of acids changed in the absorption spectra in MeOH, where a blue-shift of 8–15 nm (829–1741 cm^−1^) and a higher molar extinction coefficient (*ε*) were observed in MeOH (Table 1). From these results, compounds are supposed to exist as fairly strong aggregates, showing weak interactions with solvent in both non-polar dioxane and highly polar DMSO, and slightly higher interactions in MeOH solution.

The photophysical data of ATAs **1a**,**b** and their salts **2a**,**b** in 1,4-dioxane, DMSO and methanol (MeOH) solvents were also calculated at the B3LYP/6-31+G(d,p) level. Several singlet-A bands appear in the 260–470 nm range of the absorption spectra, but only the strongest one, corresponding to the first HOMO→LUMO transition (76→77 in our molecules), was considered, and its characteristic values are included in Table 2. It was found that the dipole moment of the acid **1a** is smaller than the dipole moment of the acid **1b** in the GS, but becomes larger in the ESs. However, the change in the polarity of salts **2a**,**b** shows the opposite trend. In the GS, they have great values that decrease by about 3-fold in ESs.

A graph comparing the shape and position of the strongest calculated band in the molecules under study is shown in Figure 3. Salts **2a**,**b** present a noticeably higher absorbance in accordance with that observed experimentally, as shown in Figure 2a. In general, a good accordance was found between the calculated and experimental values, and a linear relationship could be established between some of them, as shown in Figure 4.

The calculated wavelengths always appear between 25–50 nm higher than the experimental ones because, in their calculation, solvent molecules are considered in implicit form, which is a simplification but makes the computations much easier. This discrepancy notably increases in the calculated values in MeOH. This can be interpreted because, experimentally, the OH groups of the MeOH solvent form intermolecular H-bonds with the nitrogen atoms of the triazole ring, which is one of the main groups involved in the conjugation process; however, these H-bonds were not considered in the Theoretical Self-Consistent Reaction Field (SCRF) model that was used. This feature leads to MeOH wavelengths being caluclated from the established linear relationship with the experimental values in DMSO, Figure 4, during both the absorption and emission processes. In addition, the worst calculated values in MeOH lead to similar absorption maxima λ_01_ in DMSO and MeOH solvents (Δλ = 1.0 nm) in acid **1a**, as well as in **1b**, in contrast to those observed experimentally. The values in 1,4-dioxane also appear from this linear relationship, especially in **1a**, as they have the same λ_max_ value that can be observed experimentally in **1b**.

Calculations carried out using different DFT methods, such as the ωB97XD/aug-cc-pVTZ level reported in related molecules [18], lead to similar values and trends in the solvent as with B3LYP/6-31+G(d,p) level, confirming our results.

The experimental emission spectra showed the same trend: that the absorption spectra, although seeming to be more sensitive to the nature of the solvent, which may be due to an increase in the dipole moment value, were determined by means of quantum chemical calculations (Table 2). Thus, the emission maxima were in the 406–440 nm range and red-shifted from the nonpolar 1,4-dioxane to polar DMSO by 11 nm (625 cm^−1^) for **1a** and 19 nm (1482 cm^−1^) for **1b**, and, in contrast, were blue-shifted in MeOH (Table 1). The largest quantum yields (QYs) were obtained in nonpolar 1,4-dioxane, indicating that the lowest energy dissipation in this solvent was caused by the weakening of the solute–solvent interaction.

The peprotonation of the COOH group favors electrostatic repulsion between the salt molecules and, therefore, leads to good solubility in water; however, the formation of excimers or assembles is difficult. This feature leads to the differences in their optical behavior. In fact, the sodium salts **2a**,**b** maxima were blue-shifted in both, their absorption and emission spectra, accompained by a large decrease in their QYs, by 7.2- and 1.7-fold for **2a** and **2b**, respectively, compared to the corresponding acids (Table 1). The presence of an electron-withdrawing substituent in compounds **2b** leads to a red-shift of the emission maxima by 16 nm (890 cm^−1^) and a 4-fold increase in QY. The slight increase in the Stokes shift in the acids in polar DMSO by 17–19 nm (1025–1154 cm^−1^) confirms positive solvatochromism and partial intramolecular charge transfer (ICT). In the excited state (ES), the optimized molecular structure in the compounds under study appears to be slightly deformed compared to the ground state (GS), with a rotation of the carboxylic group and the pyrrolidine ring substituents of triazole ring. Thus, for example, in molecule **1a** in methanol, the torsional angle C11-C-C-N14 is 7.5° in GS vs. −3.5° in ES, O12=C-C-C is 13.0° in GS vs. −1.6° in ES, and N7-C8-N14-C is −1.2° in GS vs. −6.0° in ES.

The geometric difference in the GS and ES structures causes a different electronic distribution and different physical properties between both states, as can be observed in the calculated values for the emission bands listed in Table 2 and in the fluorescence spectra of Figure 3. The theoretical spectra appear to be more sensitive to the nature of the solvent than the absorbance spectra, in accordance with the experimental observations. Therefore, the linear relationship between the calculated and experimental wavelengths, shown in Figure 4, is not as good as the observations for absorbance, although a similar trend to solvent appears for some ES values. The cationic forms **2a** and **2b** fail in this relationship, perhaps because they are much more sensitive to the solvent, which does not reproduce the implicit model used.

#### 2.2.2. Concentration-Dependent Studies and Experimental Values in Mixtures with Water of Three Different Solvents

Concentration-dependent studies in organic solvents and in their mixtures with water were also carried out to shed more light on the aggregation behavior of acids **1a**,**b** in different media (*v/v* = 1/9) (Figure 5, Figure 6 and Figure 7, Table 3, and Appendix A).

The acid concentration did not affect the absorption and emission maxima in 1,4-dioxane or DMSO. By contrast, the ATA **1a** emission maximum in MeOH showed a moderate redshift by 21 nm (1101 cm^−1^), whereas acid **1b** was blue-shifted by 14 nm (1177 cm^−1^). The molar extinction coefficient (*ε*) decreased by 1.1–1.8-fold for acids and 1.8–2.2-fold for salts (Appendix A). The QY values followed the same trend and decreased, with an increase in acid concentration of 1.7–2.5-fold (Figure 5, Appendix A). This may indicate an increase in the dissipation of excitation energy due to the strengthening of intermolecular interactions and changes in the structure of aggregates or the formation of new ones. The absorption maxima of the sodium salts **2a**,**b** showed negligible or no changes in solutions with different concentrations. However, their emission maxima revealed a more remarkable red-shift in 27 nm for both salts **2a** and **2b**. The QY decreased with an increase in solute concentration of 10.6–14.2-fold for salts (Figure 5, Appendix A). Analysis of the data collected in Table 1 and Appendix A showed that even strong polar solvents, such as DMSO with a large dipole moment, slightly interfered with the optical properties of acids. Thus, the intermolecular interactions in the acid solutions are more complicated and have a stronger dependence on the solute–solute interactions than the solute–solvent interactions.

Greater effects were observed in MeOH solutions due to the stronger NCIs. This suggests the formation of complex supra-molecular structures, which is supported by the stronger intermolecular interactions that were observed compared to solute–solvent interactions. To support this hypothesis, we studied the optical properties of **1a** and **1b** at different concentrations in the presence of water (Figure 5, Figure 6 and Figure 7, Table 3, and Appendix A, Appendix A).

An apparent blue-shift in absorption by 5–24 nm (280–1644 cm^−1^) and emission of 4–17 nm (474–1076 cm^−1^) occurred when water was added to the organic solvents (Table 1 and Table 3). The largest changes were observed in the 1,4-dioxane–water mixture, and the smallest one in the methanol–water mixture. It should be noted that the absorption and emission maxima of the acids **1a**,**b** obtained in the organic solvent–water binary mixture tended toward the value of the maximum absorption or maximum emission of the corresponding salt in water (Appendix A). With an increase in water content, there was also a decrease of 2.1–2.9-fold in QY (Figure 6, Appendix A). Moreover, their values were closer to those of the salts (Figure 7b). These findings suggested that solvents, which form hydrogen bonds, have a strong effect on the structure of the fluorescent moiety, and therefore cause greater alterations in its photophysical properties. Furthermore, the p*K*a values for acids **1a** and **1b** suggest that they predominantly exist in ionized forms in the water medium. The optical characteristics determined by the spectral investigations clearly confirm this suggestion.

Sodium salts **2a**,**b** showed three special photophysical characteristics in water: (i) the absorption remained unchanged at different concentrations; (ii) the QY was significantly higher in water than in DMSO (up to 0.46 in **2a** and 0.8 in **2b**); (iii) QY decreased with an increase in acid concentration of 2.4–2.6-fold. Thus, sodium 2-phenyl-5-(pyrrolidin-1-yl)-2*H*-1,2,3-triazole-4-carboxylates appear as a new water-soluble fluorophore with excellent photophysical characteristics.

### 2.3. Fluorescence Lifetime

Fluorescence lifetime was measured in different solvents and their binary mixtures with water, because it can provide information about the excited states’ (ESs’) stability. The fluorescence lifetime is an intrinsic property of fluorophore and depends on its molecular structure. The intermolecular binding of fluorophores and the formation of excimers or nanoparticles affect their physicochemical environment, often leading to an increase in lifetime compared to that of single molecules. This is revealed by the different nature of the obtained lifetime profiles. Therefore, additional information about the fluorescent moieties formed in the ESs can be obtained.

Acids **1a**,**b** and salts **2a**,**b** showed a bi-exponential profile for most of the compounds (Table 4, Appendix A), which means that two fluorescent particles are responsible for the emissions. The exceptions are salt **2b** in DMSO (*c* = 5 × 10^−6^ M), acid **1a** in a 1,4-dioxane/water mixture and in MeOH (*c* = 5 × 10^−6^ M), with a small contribution of a third moiety, whose structure probably includes a solvent molecule, since their formation occurs in mixtures with high polar or protic solvents.

It should be noted that acids **1a** and **1b** presented high fluorescence lifetimes and radiative rate constants (*k*_r_) that were higher than the non-radiative process rate constant (*k*_r_) for most compounds. The best stability as acids and salts appears for DMSO with ⟨τ⟩_f_ = 4.15–6.71 ns.

Obviously, the lifetime of fluorescent particles decreases in acid solutions in methanol, or when water is added to their solutions in DMSO or 1,4-dioxane. This indicates that the interaction between protic solvents leads to a change in the fluorescent particles’ composition, causing the appearance of new ones with other photophysical and physicochemical properties.

### 2.4. NMR Spectroscopy of Acids ***1a**,**b*** in Different Solvents

NMR spectroscopy is a sensitive analysis method that can reveal the effects of microenvironments [48]. To confirm the formation of different intermolecular interactions, the ^1^H and ^13^C NMR spectra of the acids **1a**,**b** in DMSO-*d*_6_, methanol-*d*_4_, and 1,4-dioxane-*d*_8_ were recorded, and the signal positions of the key atoms were analyzed ((Table 5 and Table 6) and Appendix A). Theoretical values were also obtained to confirm the assignment and the structure of the synthezised compounds. Appendix A collect, for each hydrogen atom, the calculated and scaled ^1^H NMR chemical shifts, respectively, while Appendix A collect, for each carbon atom, the calculated and scaled ^13^C NMR chemical shifts, respectively. Table 5 and Table 6 list the ^1^H and ^13^C NMR average values, respectively, of similar atoms, which can be compared to the related experimental data. In general, a good agreement between the theoretical and experimental values was obtained.

The experimental δ_exp_ values obtained by ^13^C NMR in the acids **1a**,**b** appear slightly higher in methanol-*d*_4_ than the corresponding ones in DMSO-*d*_6_ and 1,4-dioxane-*d*_8_, with the lowest values being obtained in DMSO-*d*_6_. However, a clear tendency was not found in the ^1^H NMR values. This may be because the OH groups of methanol interact with the N7, N10 and N14 nitrogen atoms involved in the conjugation process, leading to a small decrease in the ^1^H NMR chemical shifts of the surrounding H3, H5 and H15 atoms, respectively compared to the values obtained in the nonpolar-1,4-dioxane-*d*_8_.

The COOH moiety showed the largest changes in the different solvents. This is because hydrogen atoms are much more affected by the environment surrounding the molecule and polarity of the solvent (specific and nonspecific interactions). The proton of OH13-group is shifted downfield by 2.38–2.41 ppm compared to the polar DMSO and nonpolar-1,4-dioxane due to strengthening of the weak interactions between solute and solvent molecules. In addition, the signals of the C-atoms of triazole and aromatic rings, and the C-atom of carbonyl group were shifted downfield by 0.7–2.2 ppm and 2.2 ppm, respectively, in methanol-*d*_4_. The strongest interference in the interactions between the solute molecules and excimers and aggregates was due to protic methanol. Due to the large interactions between the OH13-group with the solvent, which were not considered in the theoretical implicit solvent model that was used, the calculated chemical shifts that were obtained were not included in Table 5. Moreover, the hydroxyl proton is strongly shifted by deuterium atoms with the solvents that were used, leading to a very weak signal in the experimental ^1^H NMR spectra.

In our calculations, the H18′ hydrogen appears to be intramolecularly H-bonded to the C=O carbonyl oxygen; therefore, its ^1^H δ_cal_ value is highly solvent-dependent. This is why it was not included in Table 5. Excluding this H18′ hydrogen, as well as the carboxylic H13 hydrogen, as mentioned above, the theoretical (scaled) ^1^H NMR chemical shifts appear to be linearly related to the NBO atomic charges shown in Appendix A and Figure 8.

The represented relationship corresponds to the methanol values with molecule **1a**, but can also be well established in 1,4-dioxane and in DMSO solvents, as well as with molecule **1b**. For simplicity, only the relationship between 1,4-dioxane and molecule **1a** was included in Appendix A. This relationship also appears to be well established with the experimental ^1^H NMR chemical shifts, which are included in Figure 8 and Appendix A. It can be noted that the benzene ring hydrogens H2, H3, H5 and H6, as well as the H15 and H15′, have the closest linear relationship with the NBO charges. This can be interpreted due to the greater molecule geometry stability obtained around these atoms with the solvent.

A spectroscopic study [49] reveals that the ^1^H NMR chemical shifts can also be related to the C-H-stretching vibrations ν(C-H), as shown in Figure 9, which were theoretically scaled by the polynomic scaling equation procedure (PSE) [50] as well as experimentally. This feature confirms our assignment. For simplicity, only the values of molecule **1a** were included in this figure. In addition, the antisymmetric stretching wavenumbers found by IR were only considered for the CH_2_ and CH_3_ groups. The discrepancy found for the ν_as_(C-H15) stretching may be due to the fact that this mode appears to be strongly coupled with the ν_as_(C-H16) stretching.

The average values of experimental and scaled ^13^C NMR chemical shifts are plotted in Table 6. These values also appear to be strongly linearly related to the NBO charges on the carbon atoms, Figure 10. Two relationships can be established: one corresponding to carbon atoms with *sp*^2^ and *sp*^3^ hybridization (-CH_2_ and -CH_3_ groups), and another to *sp* hybridization (triazole, benzene and carboxylic groups); the latter shows noticeably higher ^13^C-NMR chemical shifts. Only the values of methanol and with molecule **1a** are represented, but this linearity is preserved with 1,4-dioxane and in DMSO solvents, as well as with molecule **1b**.

Finally, although theoretical and experimental ^15^N-NMR chemical shifts were not carried out in the present study, we think that the nitrogen atoms of our synthesized molecules are involved in the conjugation process responsible for their photophysical properties, and their ^15^N NMR chemical shift is more affected than the ^13^C-NMR values.

## 3. Materials and Methods

### 3.1. Materials and Methods

^1^H NMR and ^13^C NMR spectra were recorded with a Bruker Avance II (Karlsruhe, Germany) (400 MHz for ^1^H, 100 MHz for ^13^C) spectrometer. Chemical shifts are listed in parts per million (ppm) relative to TMS in ^1^H NMR and to the residual solvent signals in ^13^C as an external reference. Coupling constant (*J*) values are given in hertz (Hz). Signal splitting patterns are described as a singlet (s), doublet (d), triplet (t), quartet (q), sextet (sext), quintet (quin), multiplet (m), broad (br), doublet of doublets (dd), doublet of triplets (dt) or AA′XX′—spin system of *para*-substituted benzene with two different substituents. High-resolution mass spectra were obtained using an Agilent 1290 Infinity II high-performance liquid chromatography system equipped with a UV diode array detector and tandem quadrupole time-of-flight (Q-TOF) accurate mass detector Agilent 6545 Q-TOF LC–MS (Agilent Technologies Inc., Santa Clara, CA, USA). The Q-TOF instrument was operated with an electrospray ion source in positive ion mode. Elemental analyses were carried out using a Perkin–Elmer 2400 Series II CHNS/O analyzer (Shelton, CT, USA). Melting points were determined with a Stuart SMP3 apparatus (Staffordshire, ST15 OSA, UK).

UV-Vis absorption spectra were recorded on a Shimadzu UV-1800 spectrometer. The fluorescence of the sample solution was measured using a Hitachi F-7000 spectrophotometer (Tokyo, Japan). The absorption and emission spectra were recorded in DMSO, 1,4-dioxane, MeOH using 10.00 mm quartz cells. The excitation wavelength was absorption maxima. Atmospheric oxygen contained in solutions was not removed. The concentration of the compounds in the solution was 5.0 × 10^−5^ M and 5.0 × 10^−6^ M for absorption and fluorescence measurements, respectively. The relative fluorescence quantum yields (QY) were determined using quinine sulfate in 0.1 M H_2_SO_4_ as a standard (*Φ*_F_ = 0.546). Time-resolution study was recorded on Horiba FlouroMax 4 Spectrofluorometer (Kyoto, Japan) with Quanta-φ integrating sphere using FluorEssence 3.5 Software.

The reactions were monitored by analytical thin-layer chromatography (TLC) on aluminum-backed silica-gel plates (Sorbfil UV–254). Visualization of the components was accomplished by short-wavelength UV light (254 nm). Solvents were dried and distilled according to common procedures. All solvents were of spectroscopic grade. 2-aryl-5-(pyrrolidin-1-yl)-2*H*-1,2,3-triazole-4-carbonitriles **3a**,**b** were obtained by the previously reported procedures [12,16].

### 3.2. Theoretical Calculations

A conformational analysis of compounds **1a** and **1b** was carried out using DFT calculations at the B3LYP/6-311+G(2d,p) level with the GAUSSIAN 16 program package [51]. This combination of DFT method and base set has provided accurate geometries for organic molecules at a reasonably low computational cost, and is especially useful for NMR calculations. The UNIX version with the standard parameters of this package has been running in the alpha-supercomputer “Brigit” at the Computer Center of the Complutense University of Madrid. The most optimized conformer of **1a** and **1b** is plotted in Figure 1, labeling their hydrogen and carbon atoms. The frequency calculation confirms that the optimized geometry obtained for these conformers corresponds to a true minimum.

The solvent molecules were considered in implicit form using a numerical Self-Consistent Reaction Field (SCRF) model for ground and excited states, specifically the Polarizable Continuum Model (PCM) with a cavity type of Scaled VdW (Alpha = 1.100), and the polarization charges of spherical gaussians with point-specific exponents (Izeta = 3).

UV-Vis spectra were calculated in different solvents using the Time-Dependent Density Functional Theory (TD-DFT) at the B3LYP/6-31+G(d,p) level. For comparison purposes, the ωB97XD functional from Head-Gordon and coworkers [52], which includes empirical dispersion, was also used in few molecules with the aug-cc-pVTZ basis set, although it requires more computer time and memory.

^1^H- and ^13^C-NMR calculations were carried out on these conformers at the mPW1PW91/6-311+G(2d,p)//B3LYP/6-311+G(2d,p) level using the Gauge Independent Atomic Orbital (GIAO) method [53], as GIAO has become the method of choice for most NMR calculations [54]. This DFT method was selected because it provides accurate NMR chemical shifts in other molecules [55,56,57,58,59]. Moreover, a statistical evaluation of the data indicates that the most accurate prediction of ^13^C chemical shifts is achieved with this method [60] and scaling equations are made available with it [61]. GaussView 6.0 graphical interface [62] was also used because it directly converts the calculated NMR isotropic shielding tensors into chemical shifts using the tetramethyl silane (TMS) as a reference shielding. This graphic program inserted the TMS value calculated by the GIAO method at two levels: B3LYP/6-311+G(2d,p) and HF/6-31G(d). We used the B3LYP/6-311+G(2d,p)-level option because all our compounds were previously optimized at this level. The TMS values used by the program at this level were 31.8821 ppm for ^1^H and 182.466 ppm for ^13^C. With this GaussView 6.0 option, the program automatically provides the values of the δ_calc_ chemical shifts.

The chemical-shift calculations of our compounds in DMSO, 1,4-dioxane and methanol solvents were carried out using the PCM model. This model was used because it leads to an improvement in the calculated chemical shifts, especially for proton NMR, compared to other methods [61]. These values (in ppm) were corrected using the following scaling equations for ^1^H and ^13^C spectra:δ_scaled 1H_ = 0.084 + 0.9199 δ_calculated 1H_    for DMSO
δ_scaled 1H_ = 0.191 + 0.9139 δ_calculated 1H_    for 1,4-dioxane
δ_scaled 1H_ = 0.168 + 0.9084 δ_calculated 1H_    for methanol
δ_scaled 13C_ = 4.100 + 0.9728 δ_calculated 13C_    for DMSO
δ_scaled 13C_ = 4.907 + 0.9740 δ_calculated 13C_    for 1,4-dioxane
δ_scaled 13C_ = 5.028 + 0.9772 δ_calculated 13C_    for methanol

The use of empirical scaling (single-scale factor, linear scaling factor or quadratic scaling factor) is one of the most common procedures [54,61,63,64,65,66,67,68,69,70], and perhaps the most general approach to reducing the error associated with the calculation of the ^1^H and ^13^C chemical shifts, which can have average errors of up to 0.4 ppm or more for ^1^H shifts and up to 10 ppm or more for ^13^C shifts [55,66], even when using some of the best computational methods [54]. These errors are clearly too large for many applications. With the benefit of empirical scaling, namely, the application of corrections derived from linear regression procedures [56,71], the average errors can drop to below 0.1 ppm for ^1^H and two ppm for ^13^C for the same systems [60,66,67]. For many purposes, this improvement is sufficient to allow for the successful application of computed chemical shifts. The major benefit to this analysis is that the slope can be used as a scaling factor to correct the computed chemical shifts in systematic error from sources such as solvation effects, rovibratory effects, and other method limitations, all at once [60]. While the above empirical scaling approaches may be “impure” from a theoretical perspective, their usefulness is undeniable [54]. Without empirical scaling, this same degree of improvement might only be realized through the use of very high levels of theory and elaborate efforts to explicitly account for solvation, rovibratory effects, electron correlation, and other sources of error, techniques that are not readily accessible for many systems. A library of scaling factors for gas and solution phase is available to stimulate the applications of DFT NMR chemical shift calculations to resolvestructural chemistry problems [61,65].

## 4. Conclusions

In the present work, we synthesized novel sodium 2-phenyl-5-(pyrrolidin-1-yl)-2*H*-1,2,3-triazole-4-carboxylates that are effective, water-soluble blue fluorophores. The optical properties of the new heterocyclic triazole acids and their salts under various conditions (dissolved in organic solvents, in water and at different concentrations) were studied theoretically and experimentally. The most important findings of this study were as follows:(1)The investigated acids were very weak acids (p*K*_a_ = 8.5 ± 0.1 for **1a** and p*K*_a_ = 7.3 ± 0.2 for **1b**).(2)1,2,3-triazole acids appeared as stable intermolecular associates in both polar (DMSO) and non-polar (1,4-dioxane) conditions. Thus, these fluorescent nanoaggregates were responsible for their excellent photophysical properties in organic solvents.(3)The nanoaggregate stability was comparable to the strength of hydrogen bonds, since only methanol and water could affect their intra- and intermolecular bonds.(4)The optical properties of these acids in a solvent–water mixture indicated the partly destruction of the associated systems and of the anions formed.(5)A noticeable increase in the polarity of acids **1a**,**b** was revealed during the absorption of a light quantum and the transition to an excited state.(6)Using DFT methods, the calculated absorbance and fluorescence values of ATAs in different solvents were in good accordance with those found experimentally.(7)The calculated ^1^H and ^13^C chemical shifts in different solvents were scaled using different scaling equations. Their values were in good agreement with what was observed experimentally.(8)Several new relationships were established: (i) between the calculated and experimental wavelengths (nm) of the absorption and fluorescence spectra of the ATAs **1a**,**b** and salts **2a**,**b** in different solvents; (ii) between the theoretical/experimental ^1^H (^13^C-NMR) chemical shifts of compound **1a** in methanol and the calculated NBO atomic charges; (iii) between the theoretically scaled ^1^H chemical shifts and the theoretically scaled ν(C-H) stretching vibrations; (iv) between the experimental ^1^H chemical shifts and the experimental ν(C-H) stretchings.(9)The experimentally obtained spectral and calculated data showed that, along with the expected active participation of the COOH group, the nitrogen atoms of the 1,2,3-triazole ring are active participants in intermolecular interactions.

All these findings showed that the excellent photophysical properties of 2-aryl-1,2,3-triazole acids **1a**,**b** are substantially dependent on the environment; therefore, they are good candidates as selective sensors for the recognition of analytes with labile protons, especially enzymes and receptors in biological systems. 

## Data Availability

The data supporting information are available at: https://www.mdpi.com/article/10.3390/ijms24108947/s1.

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
