# Peer review of "Combined NMR Spectroscopy and Quantum-Chemical Calculations in Fluorescent 1,2,3-Triazole-4-carboxylic Acids Fine Structures Analysis"

_ijms, 2023, doi:10.3390/ijms24108947_

Round 1

Reviewer 1 Report

The authors of this manuscript presented the fine structure analysis of fluorescent 1,2,3- triazole-4-carboxylic acids with NMR spectroscopy and the quantum chemical calculations. The purpose of this study is to understand the non-covalent interactions  (NCI) of these acids and their sodium salts in different solvents and show the influence of optical properties and their biological potential. To achieve the purpose the author did the spectroscopic study and further validate the data with theoretical calculation which used TDDFT studies with B3LYP/6-31+G(d,p) level of theory. The experimental and theoretical data looks good and they are matching well in agreement. However, the outcome/findings from this study needs some stronger conclusions. I found this article is majorly missing the immediate significance towards scientific community,  the author needs to be addressed this with proper scientific manner. This article can be published after the author addressed the following comments.

1.     The absorption and fluorescence spectra of ATAs were taken in polar (DMSO), nonpolar (1,4-dioxane), and polar protic (methanol) solvents. What are the reasons to choose these set of solvents?

2.     In page 7 the author mentioned that “Introduction of an electron-withdrawing substituent in compounds 2 leads to a red-shift of the emission maxima in compound 2b by 16 nm (890 cm-1) and a 4-fold increase in QY. The slight increase in the Stokes shift of the acids in polar DMSO by 17–19 nm (1025-1154 cm-1) confirms a small positive degree of solvatochromism and negligible intramolecular charge transfer (ICT). ” This statement needs further justification. How can the  author concludes the negligible ICT? Is that predicted from theoretical data? If so what is the rate of ICT?

3.     Writing of this paper needs to be improved, e.g.

a.     In abstract the author introduced a acronym GIAO. This term needs to be addressed here.

b.     In the introduction the author mentioned about heterocyclic chemistry. What heterocyclic chemistry is important study? This needs to be addressed

c.     In page 6, the author mentioned that “The calculated wavelengths always appear between 25-50 nm higher than the experimental ones because our simplicity model uses the solvent molecules in implicit form” – what is the simplicity of the model they are referring to?

d.     In conclusion the author mentioned “Theoretical calculations confirm that found experimentally.” – what is “that” refers here?

Author Response

It is sent in the enclosed attachment

Reviewer 2 Report

The manuscript entitled “Combined NMR spectroscopy and quantum-chemical calculations in fluorscent 1,2,3-triazole-4-carboxylic acids fine structures analysis” by Safronov et al. reports on the physical-chemical analysis of the 2-aryl-1,2,3-triazole acids and their sodium salts dissolved in different solvents. The main question addressed by the research is the dependence of the photophysical properties of the mentioned compounds on chemical environment, since, being highly sensitive to the surrounding media, these appear to be attractive fluorophores that can be of a specific interest in medicine, biochemistry and materials sciences. In this respect, one can say that the presented research is relevant to the special issue “Rational Design and Synthesis of Bioactive Molecules” of the section “Molecular Pharmacology” of IJMS. However, as to whether the topic of the research is original or not, I would not be absolutely positive. If I’ve understood it correctly from the previous studies of the authors, the paper “2-Aryl-5-amino-1,2,3-triazoles: New effective blue-emitting fluorophores” published in the journal “Dyes and Pigments” (https://doi.org/10.1016/j.dyepig.2016.08.015) reports on the synthesis of a new series of 2-aryl-5-amino-1,2,3-triazole derivatives together with the studies of their photochemical properties using both experimental measurements and theoretical models. That study revealed high sensitivity of the observed fluorescent properties on the structural factors and solvent effects. The authors mentioned that 2-aryltriazolic acids 1a, b, considered in their current research, were obtained by hydrolysis from the 5-amino-2-aryl-1,2,3-triazol-4-carbonitriles, as was described in their previous paper in “Dyes and Pigments”, and then they were transformed to the sodium salts of 2-aryl-1,2,3-triazoles 2a, b by treatment with NaOH. So, what is the conceptual difference between the current work and the previous one? Any new physical-chemical properties were discovered? Any new means of physical-chemical analysis were developed? Any new reactions were presented? Any new theoretical models were proposed? Or, maybe, a specific gap was filled that was not addressed in the previous study? For now, based on what was announced in the previous paper, the present study seems to be a routine continuation of the previous one, with the only difference in studied compounds. To conclude on that, the authors should substantiate what their work adds to the subject area compared to the other published material and, specifically, to their own previous research. If the authors would fail on that point, I will suggest the authors to send their paper in some other journal, with narrower scope.

 As to specific improvements and questions, I have only a few:

1. What particular type of the SCRF model has been used?

2. What assumptions lie under the theoretical model mPW1PW91/6-311+G(2d,p)//B3LYP/6-311+G(2d,p)? Based on what it was chosen?

3. The authors write that the computations of δcalc (1H) and (13C)-NMR chemical shifts were performed at the mPW1PW91/6-311+G(2d,p)//B3LYP/6-311+G(2d,p) level by the Gauge Independent Atomic Orbital (GIAO) method. Then, the authors state that the values obtained were corrected using the tetramethyl silane (TMS) as reference shielding, with values at this computational level of 31.8821 ppm for 1H and 182.466 ppm for 13C. I vaguely suspect that the authors have meant that they used simplified IUPAC formulae and calculated chemical shifts as the differences between the shielding constants of the reference compound TMS and those of a sample. Apparently due to low-quality English, the authors’ statement is understood as if the proton and carbon δcalc were corrected with the eponymous TMS shieldings. This is nonsense.

4. In the following, the authors write: “The calculation of the chemical shifts in DMSO, 1,4-dioxane and methanol solvents was carried out under an implicit solvent model using the SCRF method." First of all, what do the authors mean by the "implicit solvent model," which was used within the SCRF method? Has the solvent been modeled as a dielectric continuum with a dielectric constant that fills the space outside the quantum system? Or, have the authors meant something else?

5. Then, the authors write "Finally, the values (in ppm) were corrected using the following scaling equations for 1H and 13C spectra.” For what reason the authors have decided to scale their calculated shifts against the experimental data? This is a trick which is used either to convert one set of data to the scale of another set of data by mapping them one into the other and deducing a mapping rule (this represents one of the ways of transfer the NMR shieldings into chemical shifts and vice versa in the case when the calculation of the shielding constant of a reference compound imposes difficulties or there is a problem with taking into account solvent effects) or to implicitly estimate the performance of computational models, as the better is the correlation, the lesser would the scaled calculated values diverge from the reference data. In the current situation, I do not see any specific reason to “fix” the calculated chemical shifts with the aid of the experimental shifts and then to compare the fixed values with the latter. Such “scaled” values may indeed be closer to the experimental values, if the correlation between the computed and experimental shifts is good, but they do not reflect a true performance of the computational model used, and, if one would apply this model with the obtained scaling factors to the other set of compounds, the predictions of chemical shifts would hardly be reliable, to say the least. Thus, I do not see any value in the obtained scaling factors. To my mind, in the context of this work, it’s just a shifty fitting of the calculated values to the experiment. In this context, what data are presented in the Tables 5, 6? There is no mentioning about the scaling in the NMR section.  

6. I do not quite follow what particular information about the non-covalent interactions have the authors obtained from their experimental measurments and theoretical quantum chemical calculations? The most interesting and most problematic is the analysis of the contributions from different types non-covalent interactions (van der Waals interactions, hydrogen bonds, electrostatic interactions, supramolecular hyperconjugation) to the total binding energy. No analysis of the importance of these particular effects on the binding energy was presented. Moreover, in the theoretical modeling, the authors used the SCRF model (accounting for the electrostatic effects, at most) without explicit solute-solvent interactions. How the authors intended to investigate the hydrogen bond formation in the case of MeOH based only on the SCRF model? The AIM analysis should be carried out in real supramolecular complexes.

The conclusions are consistent with the evidences and arguments presented, but they do not address the main question posed in the beginning. There was no thorough high-quality theoretical analysis of the non-covalent interactions performed. The references are appropriate. English requires extensive stylistic correction. On the tables and figures I have no specific comments (the quality is fine).

 To summarize, I recommend a major revision of the manuscript. 

English requires extensive stylistic correction.

Author Response

It is sent in the enclosed attachment

Round 2

Reviewer 2 Report

The authors have tried to answer my comments. I appreciate that. My main concern about the novelty has been fulfilled. Though, as to my other requests about the theoretical approaches used and especially about the AIM analysis, I see no acceptable explanations. However, I understand that the work is purely synthetic combined with physicochemical analysis, and the quantum-chemical calculation is not its forte. In particular, the authors, apparently, did not understand my remark on the scaling factors and they do not actually see the difference between the cases when the linear regression makes sense and that when it does not.  A negligent attitude to the theoretical part is upsetting, because the work is good and a real thorough theoretical analysis, especially, that of the non-covalent interactions, would represent a fine complement to this work. In general, I do not see any obstacles not to recommend this work for publishing. Next time, I expect more vigilance from “learned” authors regarding their quantum-chemical calculations.